



# On the comparison of strain measurements from fibre optics with dense seismometer array at Etna volcano (Italy)

Gilda Currenti[1], Philippe Jousset[2], Rosalba Napoli[1], Charlotte Krawczyk[2,3], Michael Weber[2,4]

5  [1]Istituto Nazionale di Geofisica e Vulcanologia-Osservatorio Etneo, Piazza Roma 2, Catania, Italy
[2]GFZ German Research Centre for Geosciences, Telegrafenberg, Potsdam 14473, Germany
[3]Technical University Berlin, Ernst-Reuter-Platz 1, Berlin10587, Germany
[4]University Potsdam, Karl-Liebknecht-Str. 24-25, Potsdam 14476, Germany

10  *Correspondence to*: Gilda Currenti (gilda.currenti@ingv.it)

**Abstract.** We demonstrate the capability of Distributed Acoustic Sensing (DAS) in recording volcano related dynamic strain at Etna (Italy). In summer 2019, we gathered DAS measurements from a 1.5 km long fibre in a shallow trench and seismic records from a conventional dense array comprising 26 broadband sensors, deployed in Piano delle Concazze close to the summit area. The multifaceted style of Etna activity during the acquisition period gives the extraordinary opportunity to record and detect tiny strain changes (few $10^{-8}$ strain) in correspondence with volcanic events. To validate the DAS strain measures, we explored array-derived methods to estimate strain changes from the seismic signals and to compare with strain DAS signals. A general good agreement was found between array-derived strain and DAS measures along the fibre optic cable. Short wavelength discrepancies correspond with fault zones, showing the potential of DAS in mapping local perturbations of the strain field, and thus site effect, due to small-scale heterogeneities in volcanic settings.

20  **1 Introduction**

In the recent past, direct measures of the strain field have been hampered by the complex installation and high maintenance cost of traditional strainmeters. In the best case, few instruments are deployed in borehole, thus sensing the strain in only few points (Currenti and Bonaccorso, 2019). Numerical investigations have clearly shown that seismic velocities, gradient displacements and strain dramatically changes at sharp boundaries and in presence of steep topography (Kumagai et al., 2011; Jousset et al., 2004; Cao and Mavroeidis, 2019). This poses challenges in interpreting accurately strain observations relying on only few measurements points.

Nowadays advances in the direct measurement of strain at an unprecedented high spatial and temporal sampling over a broad frequency range have become possible due to the growing use of Distributed Acoustic Sensing (DAS) technology. Since the application of DAS in geoscience is an emerging field, many open questions still exist on the DAS device response and the coupling effect between the fibre and the ground, that strongly depends on conditions of fibre installation. A few field



experiments in various environments have been designed to compare DAS strain measures and indirect strain estimates from collocated or nearby traditional sensors, such as geophones and broadband seismometers (Jousset et al., 2018; Wang et al., 2018; Yu et al., 2019; Lindsey et al., 2020).

In this paper we compare array-derived strain with direct DAS strain measures, acquired at Etna volcano (Italy). As test site we selected Etna for its multifaceted and frequent activity. Lava flows, explosive eruptions with ash plumes, and strombolian lava fountains commonly occur from Etna summit craters and these multiple episodes of different style results in a wide variety of signals that we detected and investigated. A dense seismic array and a 1.5 km long fibre optic cable connected to a DAS interrogator were jointly installed few kilometers away from the summit craters to assess the reliability of DAS in recordings volcano related strain changes. To validate the DAS measures, we explore several methods for the indirect estimates of strain field from dense seismic array data.

Several methods exist in literature for estimating strain from high to quasi-static frequencies relevant for seismologic and geodetic investigations. Those include the evaluation of seismic rotational components (Basu et al., 2013), the analysis of the performance of rotational seismometers (Suryanto et al., 2006), the estimation of strain-meter response and calibration (Donner et al., 2017; Currenti et al., 2017), the computation of strain rate maps (Teza et al., 2004; Shen et al., 2015), the estimation of the stress field induced by the passage of seismic waves (Spudich et al., 1995), the determination of seismic phase velocity (Gomberg and Agnew, 1996) and constraints on seismic sources (Donner et al., 2017; Spudich and Fletcher, 2008; Langston and Liang, 2008).

In general, those methods can be grouped in two main families relying on single or multiple station methods. On one hand, the single station method, pertinent only to seismology, has been widely used to estimate dynamic gradient displacements and strain tensor from the translational components of a 3C seismometer (Gomberg and Agnew, 1996). It assumes that seismic energy is carried by plane waves with known horizontal velocity in a laterally homogeneous medium. On the other hand, multiple station procedures involve the use of displacements or velocity recordings from a number of close sensors or dense arrays. In such a case, the measurements are processed using spatial interpolation approaches (Sandwell, 1987; Paolucci and Smerzini, 2008; Sandwell and Wessel, 2016) or the well-known seismo-geodetic method (Spudich et al., 1995; Shen et al., 2015; Teza et al. 2004; Basu et al., 2013). Beside the use of geostatistical approaches, generally applied for optimal interpolation of a discrete scalar field, several interpolation schemes have been formulated using the solutions of suitable elasticity problems (Sandwell, 1987; Wessel and Bercovici, 1998; Paolucci and Smerzini, 2008). A complete review of these latter methods can be found in Sandwell and Wessel (2016). In the same paper the Authors, following the biharmonic spline method (Sandwell, 1987), proposed a new solution, in which the interpolation functions originate from the computations of the analytic Green's solutions of an elastic body subjected to in-plane forces. Finally, the seismo-geodetic method is based on the Taylor expansions of the displacements field of first (Spudich et al., 1992) or higher order derivatives (Basu et al., 2013). Both interpolation and seismo-geodetic methods have been used for seismic dynamic and geodetic quasi-static strain estimates. While the single station procedure estimates the strain at the single position of the collocated sensor, the other two methods allow deriving a map of strain on irregular grid points by making use of sensor array measurements. This makes the multiple



station approach appropriate to estimate strain using dense seismic array methods and compare it with values derived from
        DAS along fibre cable.

        Here, we extend the interpolation method proposed by Sandwell and Wessel (2016) and use the seismo-geodetic method in
        the formulation by Shen et al. (2015) in order to derive strain estimates from the dense seismic array data acquired at Etna.
        The aim of the paper is twofold: (i) exploring the performance of interpolation and seismo-geodetic methods in deriving the
strain field along a fibre optic cable and (ii) validating the DAS measurements acquired during the experiment at Etna volcano.

## 2 Data set

We designed the experiment in Piano delle Concazze at the Etna summit (Fig. 1) in order to test the potential of the DAS
technology in a volcano-seismology application.

Deployment of equipment on active and dangerous volcanoes is challenging due to harsh environment and danger associated
with the volcanic activity. Piano delle Concazze is a large flat area (elevation of about 2800 m), on the northern upper flank of
        the Etna volcano dominated by the North East Crater. It is bounded by the upper extremity of the North-East Rift, a preferential
        pathway for magma intrusions due to its structural weakness (Andronico and Lodato, 2005), and by the rim of the depression
        of the Valle del Leone. The area is affected by several North South-trending faults, that result from the accommodation of the
        extension exerted by the North-East Rift (Azzaro et al., 2012; Napoli et al., in print). Therefore, Piano delle Concazze is an
area where new methodology can be tested, close enough to the active craters to study volcanic processes (ca. 1.8 km away
        from the Etna summit craters) and yet safe.

        In order to record strain changes related to volcanic activity, we jointly deployed in Piano delle Concazze (Fig. 1):

        1.        a 1.5 km long fibre connected to a DAS interrogator ("Ella", an iDAS® from Silixa) set up in Pizzi Deneri
        Observatory. We interrogated the fibre from 1 July to 23 September 2019. A long trench was dug to deploy the fibre cable at
a depth of about 40 cm. At a sampling rate of 1 kHz the DAS acquired the strain rate along the axial direction of the cable with
        a spatial resolution of 2 m and a gauge length of 10 m. This results in a dataset of 824 channels distributed along the fibre path.
        The spatial calibration and locations of the fibre channels were determined by jumping at several points near the fibre with
        jumping locations determined by handy GPS. The position of the channels between two successive jump locations were
        computed by linear interpolation.
2.        a dense seismic array comprising 26 Trilium Compact broadband sensors (interstation distance of about 70 m),
        distributed over an area of about 0.2 km2 (Fig. 1). The sensors were placed at a depth of about 40 cm deep holes in the
        compacted pyroclastic deposit. A Cube digitizer was used to acquire the data at a sampling rate of 200 Hz.

        We designed the broadband sensor distribution and the path of the fibre optic cable, such that we could compare both records
        with several methods.





**3 Methodology**

We estimate the strain field from the seismic array data using two different algorithms. The first is a new spatial interpolation method, in which we extend the work presented in Sandwell and Wessel (2016). The second is the seismo-geodetic method in the formulation proposed by Shen et al. (1996; 2015). For both methods the description and the analysis are limited to the 2D domain, since the investigated area is almost flat. However, extension to 3D is straightforward.

**3.1 Spatial interpolation method**

In the spatial interpolation method (SIM) proposed by Sandwell and Wessel (2016), the general solution of the horizontal displacement components u,v at any location $\vec{r}$ within a N-element seismic array is given by:

$$u(\vec{r}) = \sum_{j=1}^{N} q(\vec{r} - \vec{r}_j) f_x^j + p(\vec{r} - \vec{r}_j) f_y^j$$

$$v(\vec{r}) = \sum_{j=1}^{N} w(\vec{r} - \vec{r}_j) f_x^j + p(\vec{r} - \vec{r}_j) f_y^j$$

(1)

where $q, p, w$ are the analytical Green's functions (Sandwell and Wessel, 2016), which solve the quasi-static balance equation

of an elastic body subjected to in-plane body forces $\boldsymbol{f_x, f_y}$ applied at the station location $\vec{r}_j$. Starting from these deformation solutions we derive the strain field $\boldsymbol{\varepsilon}$, which read as:

$$\varepsilon_x(\vec{r}) = \sum_{j=1}^{N} \frac{\partial q(\vec{r} - \vec{r}_j)}{\partial x} f_x^j + \frac{\partial w(\vec{r} - \vec{r}_j)}{\partial x} f_y^j$$

$$\varepsilon_{xy}(\vec{r}) = \frac{1}{2} \sum_{j=1}^{N} \left[ \frac{\partial q(\vec{r} - \vec{r}_j)}{\partial y} + \frac{\partial w(\vec{r} - \vec{r}_j)}{\partial x} \right] f_x^j + \left[ \frac{\partial w(\vec{r} - \vec{r}_j)}{\partial y} + \frac{\partial p(\vec{r} - \vec{r}_j)}{\partial x} \right] f_y^j$$

$$\varepsilon_y(\vec{r}) = \sum_{j=1}^{N} \frac{\partial w(\vec{r} - \vec{r}_j)}{\partial y} f_x^j + \frac{\partial p(\vec{r} - \vec{r}_j)}{\partial y} f_y^j$$

(2)

where the gradients of the functions $q, p, w$ are given by:

$$\frac{\partial q(r)}{\partial x} = (3-v)\frac{x}{r^2} - 2(1+v)\frac{xy^2}{r^4} \qquad \frac{\partial q(r)}{\partial y} = (3-v)\frac{y}{r^2} + 2(1+v)\frac{yr^2 - y^3}{r^4}$$

$$\frac{\partial w(r)}{\partial x} = -(1+v)\frac{yr^2 - 2x^2 y}{r^4} \qquad \frac{\partial w(r)}{\partial y} = -(1+v)\frac{xr^2 - 2xy^2}{r^4}$$

$$\frac{\partial p(r)}{\partial x} = (3-v)\frac{x}{r^2} + 2(1+v)\frac{xr^2 - x^3}{r^4} \qquad \frac{\partial p(r)}{\partial y} = (3-v)\frac{y}{r^2} - 2(1+v)\frac{x^2 y}{r^4}$$

(3)





and $\upsilon$ is the Poisson's ratio and r is the distance between the location $\vec{r}$ and the $\vec{r}_j$ station position. We determine the body

forces, applied at the N locations of the array stations, so that they match the displacement data using Eq (1). Therefore, we

solve for a set of N vector body forces $\boldsymbol{f_x}, \boldsymbol{f_y}$ with j=1…N, by setting up a 2N x 2N linear system of equations constrained

by the horizontal displacement field recorded by the broadband seismic array. The linear system is solved using a least square

method. Once the body forces are determined the components of the strain tensor can be computed at the location of the fibre

channel using Eq (2). To avoid the singularity in the computation of the Green's functions (Eqs. 3), a $\Delta r$ distance factor is

added to the distance term (Sandwell and Wessel, 2016). Therefore, the solutions depend on the values of the Poisson ratio

and the $\Delta r$ distance factor, which need to be tuned in order to obtain the optimal interpolation.

### 3.2 Seismo-geodetic method

The seismo-geodetic method (SGM) is based on the Taylor expansion of the displacement field and has been applied using

different assumptions and strategies which lead to very similar formulations (Spudich et al., 1995; Shen et al., 1996; Teza et

al., 2004; Langston and Liang, 2008; Basu et al., 2013; Langston et al., 2018). Here, we follow the formulation proposed in

Teza et al. (2004). The displacement components $u_i(x_k, y_k)$ with *i=x, y*, recorded at the *k-th* station are expanded with first

derivative terms as:

$$u_i(x_k, y_k) \approx u_i(x_P, y_P) + \frac{\partial u_i}{\partial x}(x_k - x_P) + \frac{\partial u_i}{\partial y}(y_k - y_P) \tag{4}$$

where the first term $u_i(x_P, y_P)$ is the *i-th* displacement component and the second and the third terms are the

gradient displacements at a point $P(x_p, y_p)$, respectively. By computing this expression for the N stations

of the array, a linear system of N equations is derived, $\boldsymbol{Gm = d}$, whose solution provides the

displacements and their gradient at the point P. If the displacement gradients are rewritten in terms of

symmetric and rotational strain components, the system of equations read as (Shen et al., 2015):


$$\boldsymbol{G} = \begin{bmatrix} 1 & 0 & \Delta x_1 & 0 & \Delta y_1 & \Delta y_1 \\ 0 & 1 & 0 & \Delta y_1 & \Delta x_1 & -\Delta x_1 \\ \cdots & & \cdots & & \cdots & \\ \cdots & & \cdots & & \cdots & \\ \cdots & & \cdots & & \cdots & \\ 1 & 0 & \Delta x_N & 0 & \Delta y_N & \Delta y_N \\ 0 & 1 & 0 & \Delta y_N & \Delta x_N & -\Delta x_N \end{bmatrix}$$

$$\boldsymbol{m}^T = \begin{bmatrix} u_x & u_y & \varepsilon_x & \varepsilon_y & \varepsilon_{xy} & \omega \end{bmatrix} \tag{5}$$





$$\boldsymbol{d} = \begin{bmatrix} u_x(x_1, y_1, z_1) \\ u_y(x_1, y_1, z_1) \\ \dots \\ \dots \\ u_x(x_N, y_N, z_N) \\ u_y(x_N, y_N, z_N) \end{bmatrix}$$

The vector of the unknowns m ($^{\mathrm{T}}$ indicates the transposition operator) is composed of the horizontal displacements, the strain tensor components and the rotation. The vector $\boldsymbol{d}$ is formed by the observed displacements at the stations of the array and $\Delta x_i$ and $\Delta y_i$ are the relative positions between the *i-th* station and the point P. In the first order Taylor expansion, the computed displacement gradients are constant (Basu et al., 2013; Shen et al., 2015) and, hence, the estimated strain tensor is spatially uniform over the entire extent of the array. Therefore, this method provides one representative value within the whole seismic

array. To overcome this limit, a distance-weighted approach is introduced which locally weights the data on the basis of the relative distance $R_i$ between the station and the point P (Shen et al., 2015). The diagonal terms of the weighting matrix are computed as exponential functions $W_i = \exp\left(-\frac{R_i^2}{D^2}\right)$ where $D$ is a spatial smoothing parameter. The system of equations is solved using a weighted least square approach $\boldsymbol{m} = (\boldsymbol{G}^T \boldsymbol{W} \boldsymbol{G})^{-1} \boldsymbol{G}^T \boldsymbol{W} \boldsymbol{d}$. The advantage of this method with respect to the Spudich formulation (1995) is that the displacement and the strain fields can be interpolated at any point P within the array

without the need for defining a reference station.

**3.3 Strain computations along fibre cable axial direction**

Using one of the above methods, the strain tensor is computed at all channels of the fibre and it is projected along the local fibre direction to compute the local axial strain. Both procedures are iterated at each time step to obtain the time series of dynamic strain at all the channels of the fibre using the time series of the seismic array.

The accuracy of the array-derived strain estimates is limited by the inter-station distance L. The error of the strain estimate is given by $e = 1 - \frac{\sin(\pi L/\lambda)}{\pi L/\lambda}$ where $\lambda$ is the signal wavelength (Bodin et al., 1997). Therefore, an accuracy of 90% is achievable if the dominant wavelength is larger than about four times the inter-station distance (Spudich and Fletcher, 2008 and reference therein). Considering that in Piano delle Concazze, the average apparent velocity $V_a$ is not greater than 2 km/s (Saccorotti et al., 2004), we can deduce that the strain estimates can be derived from the seismic array data with a good accuracy

up to a frequency $f_{max} = {V_a}/{4\Delta}$, i.e. of about 6 Hz, for the average station distance of 70 m..



## 4 Results

During the acquisition period, the Etna volcano activity was mainly characterized by discontinuous strombolian explosions and isolated ash emissions from most of the active summit craters (North East, New South East, Bocca Nuova and Voragine),
distant about 2 km from Piano delle Concazze (Fig. 1). These activities, occurring at fluctuating intensity, preceded and accompanied the short-lived effusive eruptions on 18 and 27 July 2019 from the New South East Crater (NSEC), and continued until the end of the experiment. A wide variety of signals have been recorded, but we focused our analysis on classes of events with a frequency content less than 6 Hz. Among the several signals here we selected two types of events (Fig. 2): (i) a volcanic explosion (VE) accompanying the strombolian activity at NSEC on 6 July 2019 and (ii) a long period event (LP) on 27 August
2019 preceding the intensification of the eruptive activity at the summit craters in early September 2019. In agreement with other similar events recorded at Etna (Cannata et al., 2009), the spectra show frequency contents in the range 0.1 to 1 and 1 to 5 Hz for the LP and the VE events, respectively. Both broadband and DAS signals are filtered with a 3th order Butterworth filter. The integration of DAS data over time provides the strain along the fibre optic cable. Data are down-sampled from 1 kHz to 200 Hz for direct comparison with strain derived from broadband array signals.

Some of the broadband seismometers (e.g. Bb02, Bb03, Bb04) are collocated, with a distance of less than 1 m with the fibre optic cable. This configuration enables also to apply the single station method, which, under the plane wave assumption, relates the DAS strain at the nearest channel with the broadband particle velocity as $\varepsilon_x = -p\dot{u}_x$, where $\varepsilon_x$, $\dot{u}_x$ and $p$ are the strain, the particle velocity and the horizontal slowness (i.e. the inverse of the apparent wave propagation velocity) along the cable axial direction $\vec{x}$, respectively. In Figure 2 we report the comparison between the DAS strain at the nearest channel, here
channel 501, and the particle velocity at the station Bb04. A horizontal slowness of about 1 s/km is estimated to optimally scale the particle velocity and match with the strain. For both events a generally good agreement is achieved (Fig. 2 b and d). This single station method is appropriate to estimate the strain only at the position where the seismometer and the cable are collocated. Because of the high spatial variability of the strain along the fibre optic cable (Fig. 2), we attempt to estimate the strain at all channels using the multiple station methods described in the previous section.

Velocity data are first integrated over time to derive the displacement field, which is then used in the SIM and SGM methods. Residuals, in terms of RMSE (Root Mean Square Error) misfits, are computed between DAS strain measurements and the strain derived from the seismic array. The RMSE values computed along the fibre gives a local estimate of their respective performance (Figs 3 and 4).

The SGM depends strongly on the spatial smoothing parameter D (see above). Therefore, we performed several computations
by varying the D parameter and selected the value at which the cumulative RMSE is minimized (Fig. 5). A minimum RMSE is found with a D value of 75 m and 100 m for the VE and LP events, respectively. Similar exploration in the parameter space was performed for the SIM, which depends on the $\Delta r$ distance factor. Here the best fit for $\Delta r$ was found for a value of 20 m and 10 m for the VE and LP event, respectively. An additional search was also carried out on the Poisson's ratio to improve the fit, with an optimal value of 0.25.





An overall good match is achieved for both methods (SGM and SIM) along the fibre. Higher RMSE misfits concentrate at the corner points, where, the cable direction turns abruptly and the axial strain is locally disturbed. Furthermore, discrepancies are also observed in regions where the fibre crosses fault zones (Figs. 3 and 4).

## 5 Discussion

We investigated several methods for indirect strain estimates from dense seismic array data aiming at the assessment of DAS
records. Because of the high spatial sampling of the DAS data, methods suitable to provide strain distribution on irregular points are promising. After a straightforward recasting and derivation of equations, we compared the SIM and the SGM methods for the estimates of strain fields from dense array data, which, in similar forms, have been applied in several fields of seismology and geodesy. To our knowledge, these approaches are adapted and tested on DAS data here for the first time.
SIM and SGM offer some advantages over single station method. Both provide a direct comparison between strain data and
their estimates from velocity data, without any assumption on the local phase seismic velocity as required by single station procedure. Particularly, when media are dispersive, the assumption of a constant phase velocity is possibly prone with errors. Usually, and especially in volcanic area, the estimate of a phase seismic velocity is challenging, because of the presence of strong heterogeneity and fractured zones. Particularly, in Piano delle Concazze, the phase seismic velocity varies markedly along the fiber cable because of the local complex structural geology that is characterized by a lava flow succession interbedded
with volcaniclastic products (Branca et al., 2011) and by sub-vertical North-South trending faults affecting the superficial layers up to a maximum depth of about 40 m (Azzaro et al., 2012; Napoli et al. in print).
Indeed, our results highlight strong discrepancies between direct DAS measures and indirect strain estimates in correspondence of fault zones (Figs. 3, 4). Thanks to the high spatial resolution of the DAS records, it was possible to observe that strain is affected by small-scale heterogeneity in the soil structure. Local strain perturbations are much shorter in wavelength (few tens
of meters) with respect to the wavelength resolution of the array ($\lambda > 200$ m). The propagation of the seismic field is perturbed by the local geology with clear amplifications of strain in correspondence of fault zones (Fig. 2). This is in agreement with the amplification of DAS strain measures observed in the proximity of faults in the DAS experiment carried out in Iceland (Jousset et al., 2018; Schantz, 2020) and at Etna (Jousset et al., submitted). These findings clearly show that complex site effects introduced by lateral heterogeneities may increase significantly the strain with respect to simplified evaluations, that may
underestimate such effects severely. These findings emphasize the potential in the use of DAS technique to both characterize and monitor local strain in the vicinity of fault zones by providing a direct measure of strain at a spatial sampling unachievable even with large-N arrays. However, local strain perturbations are not found at all the positions where the fibre cable crosses the fault systems (Figs. 3, 4). The cause of this dissimilar response of the ground is hindered from the limited geological characterization of the area. Further investigations with complementary geophysical exploration methods could help in
understanding the different behavior at different locations.





The discrepancies are higher for the shorter wavelength signal recorded during the VE event with respect to the LP event (Figs. 6, 7), for which good estimates are obtained at almost all channels (Figs. 4, 8). For the longer wavelengths (e.g. LP event), SIM and SGM (Fig. 6) perform better than the single station approach (Fig. 2). These findings conform to the estimate of the accuracy, which is dependent on the wavelength signals.

Both methods allow for estimating strain on irregular points exploiting all the available dataset instead of relying on a single point measure (Figs. 7, 8). Moreover, the derivation of the analytical strain solutions in the SIM: (i) avoids to use finite-difference scheme to derive strain from regular grid point distribution of displacements, and hence, (ii) provides the strain at any point of the investigated area once the body force coefficients have been estimated by solving the linear system of equations. This results in a greater accuracy.

Finally, the proposed approaches offer the possibility to combine seismic array data with DAS measures to derive a 2D map of the local strain of the investigated area. By recasting the system of linear equations, it is straightforward to include the DAS strain measures and perform a joint inversion.

## 6 Conclusions

The joint deployment of a DAS device and of a dense seismic array at Etna summit offers a unique opportunity to detect and
quantify accurately strain changes related to volcanic activity. The data set recorded during summer 2019 showed the great potential of distributed fibre optic sensing in volcanic environment. To our knowledge this is the first time that tiny strain changes related to volcanic explosive and LP events have been clearly detected by DAS technology opening new perspectives for its use in volcano monitoring. The high spatial sampling of DAS measures confirms the high variability of strain variations in complex geology. These findings also contribute in explaining the difficulties often encountered in interpreting local strain
changes from single strainmeter observations. The indirect strain estimates derived by the dense seismic array match quite well with the direct DAS measurements. Our findings validate both the proposed methods and the accuracy of DAS measurements in sensing strain changes produced by volcanic processes.

**Code and data availability.** MATLAB scripts and data are available upon request to the corresponding Author.

**Author contributions.** PJ, CK and MW conceived and supervised the project. PJ, GC and RN were involved with the experiment planning. GC conceptualized this study and performed the analyses. All Authors contributed to the acquisition of the field data, the writing of the manuscript and the discussion of the results.

**Competing interests.** The Authors declare no competing interests





**Acknowledgements.** Broadband seismometers and data logger equipment are from the Geophysical Instrument Pool Potsdam (GIPP). This work received funding from the GeoForschungZentrum Potsdam and the Helmholtz Association. The experiment was also financially supported through the Trans National Activity "FAME" within the EUROVOLC project (EU grant
agreement ID: 731070). Thanks are due to V. Parra and the INGV staff composed of S. Consoli, D. Contrafatto, G. La Rocca, D. Pellegrino, M. Pulvirenti for their great help in the field work.

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





**Figure 1:** Digital terrain model of Piano delle Concazze in the North-East flank of Etna volcano (Palaseanu-Lovejoy et al., 2020) with the dense broadband seismic array (26 stations; Bb1-Bb26) and the Distributed Acoustic Sensing (DAS) cable layout. No data were recorded by the station Bb05 because of technical problem. The DAS interrogator was hosted inside the Pizzi Deneri Observatory, that is about 2 km away from the five summit craters (North-East: NEC, Voragine: VOR, Bocca Nuova: BN, South-East: SEC, New South-East: NSEC). The investigated area, almost flat, is crossed by a sub-vertical fault system (redrawn after Azzaro et al., 2012). Geographic coordinates (in km) are in UTM33S system.



 **Figure 2:** Time series of DAS strain during a small volcanic explosion (VE) at Etna on 6 July 2019 (a, b) and during a long-period (LP) event on 27 August 2019 (c, d). The DAS strain is computed by integrating over time the DAS records. Data are filtered in the frequency range 1 to 5 Hz for the VE event (a, b) and between 0.1 to 1 Hz for the LP event (c, d). (b, c) The broadband velocity at station Bb04 (black line in b and d), located in proximity of the DAS channel 501 (red line), are projected along the cable direction to compare the data using the single station procedure. An average scaling factor of 1000 m/s for the slowness value is found in both cases.







**Figure 3:** RMSE residuals between DAS strain measurements (Fig. 2a) and the strain derived from the seismic array along the fiber for the VE event. (Top) DAS vs. the seismo-geodetic method (SGM). (Bottom) DAS vs. the spatial interpolation method (SIM). Black lines and open circles indicate faults and seismometers shown in Fig.1, respectively. For more details see text.






**Figure 4:** RMSE residuals between DAS strain measurements (Fig. 2c) and the strain derived from the seismic array along the fibre for the LP event.


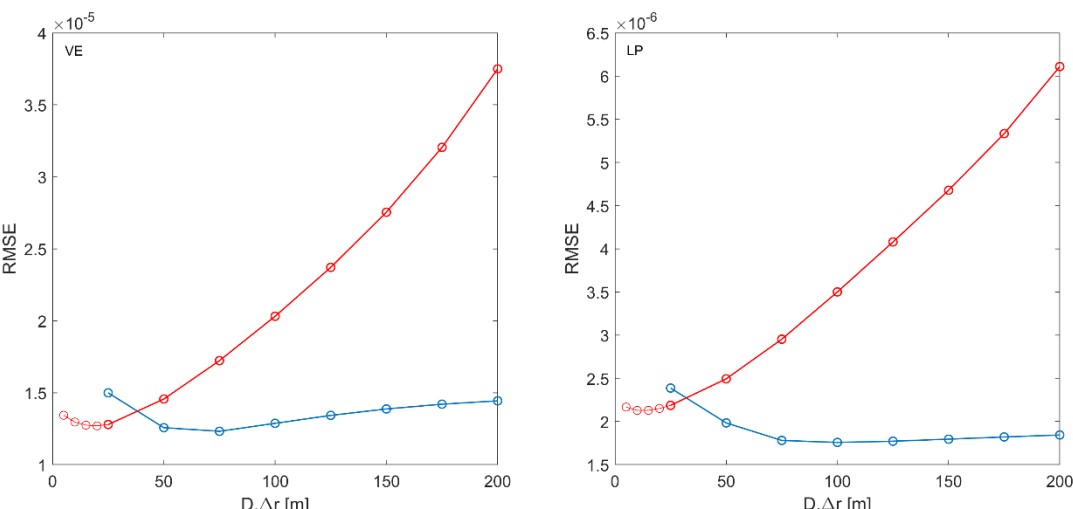

**Figure 5:** Cumulative residuals over all fiber channels between DAS strain measurements and strain derived from SGM (blue line) and from SIM (red line) for different smoothing parameters (D for SGM and Δr for SIM, respectively). Computations are performed for the VE (left) and LP (right) events, respectively.

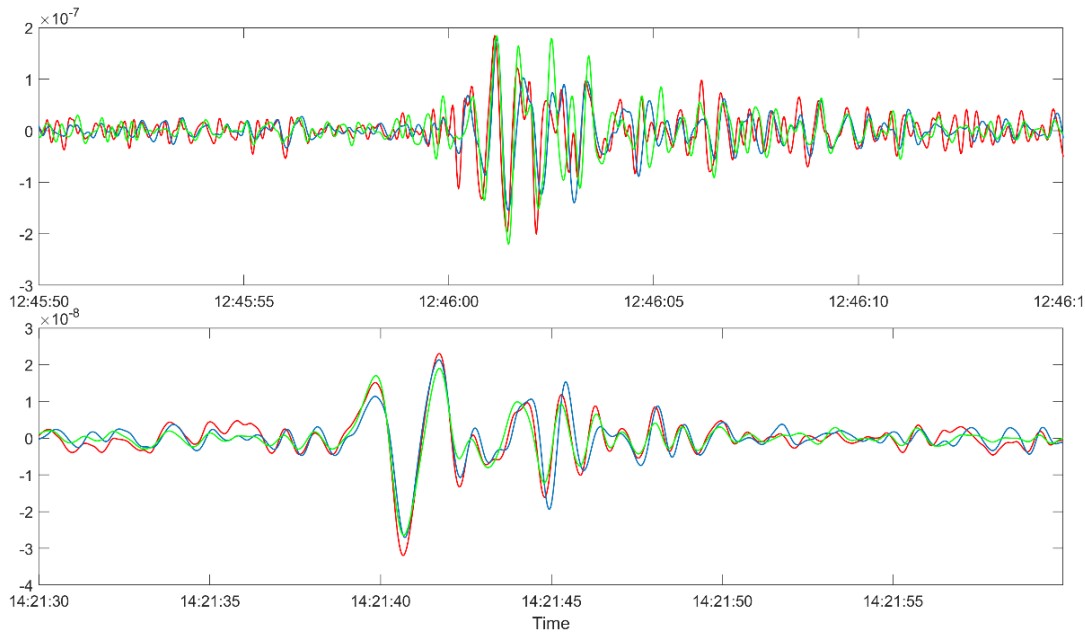

**Figure 6:** Array-derived strain estimates at station Bb04 (Fig. 1) using the SIM (green line) and the SGM (blue line) in comparison with DAS strain data (red line). Computations are performed for the VE (top) and LP (bottom) events.



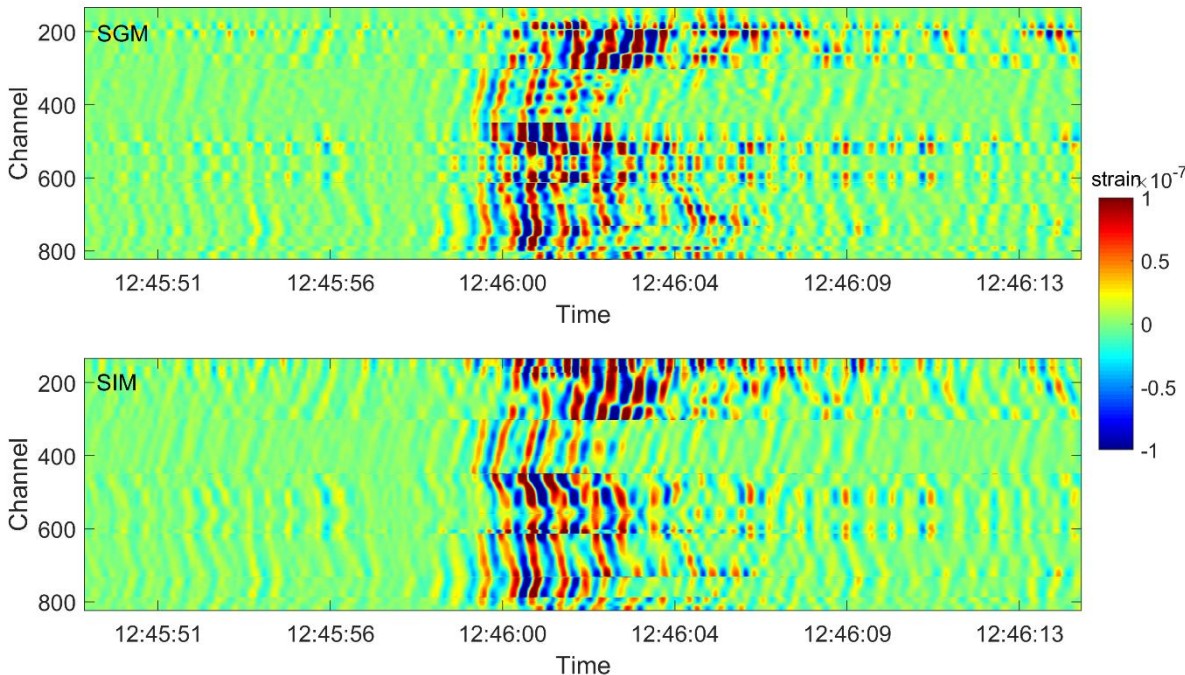

**Figure 7:** – Strain estimates over all fibre channels using SGM (top) and SIM (bottom) for the VE event. The solutions with the minimum RMSE are chosen, respectively (Fig. 5).






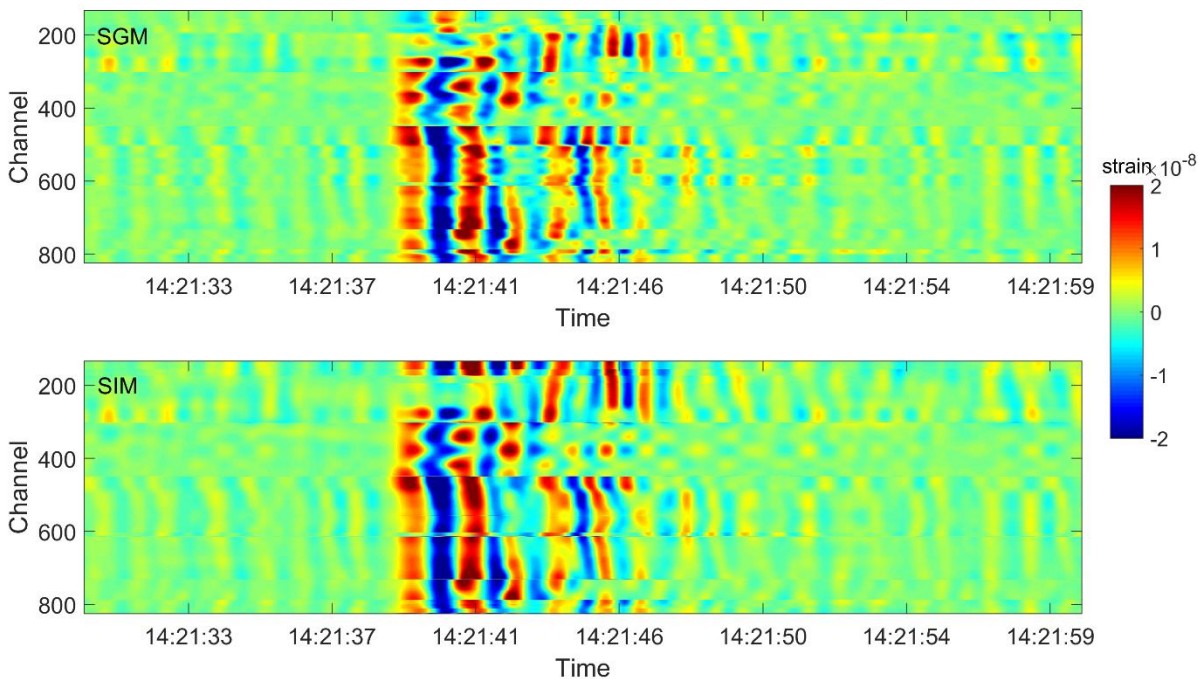

**Figure 8:** Strain estimates over all fibre channels using SGM (top) and SIM (bottom) for the LP event. The solutions with the minimum RMSE are chosen, respectively (Fig. 5).
