# Peer review of "On the comparison of strain measurements from fibre optics with dense seismometer array at Etna volcano (Italy)"

_Solid Earth, 2020_

## Author Comment (AC1)

**Manuscript Ref. No. se-2020-216:** "On the comparison of strain measurements from fibre optics with dense seismometer array at Etna volcano (Italy)" by Gilda Currenti et al.

**Anonymous Referee #1 (General Comments):**

Distributed Acoustic Sensing (DAS) is a technique to measure strain (or strain rate) along a fiber optic cable at an unprecedented spatial resolution and is now used even in seismology. One concern of DAS is fidelity in the absolute amplitude of measured strain attributed to the coupling between the ground and the cable. In this study, based on a laborious seismic array observation at Etna volcano, the authors estimated strain by interpolating the seismic wave field with two different methods: the spatial interpolation methods and the seismo-geodetic method. Comparing these seismically estimated strain and DAS strain, the authors showed that both strains agreed well, which shows the fidelity in amplitude of DAS observation. This result made an essential contribution to validating seismological applications of DAS. And I highly evaluate the elaborate array observation by the authors in such a high mountain. The manuscript is clearly written. The subject of this manuscript is up-to-date and suitable for this journal. I would suggest this manuscript for a minor revision. The following comments should be addressed in the revised manuscript.

**Authors:** We are glad that the main message of the manuscript was well appreciated. We really thank the referee for the positive criticism and detailed questions which stimulate us in deepening the analysis and improving the manuscript.

**Anonymous Referee #1 (Specific comments):**

The two interpolation methods introduced different smoothing parameters. I am curious how these smoothing parameters are related to the gauge length of DAS. Can the authors comment on that? **Authors:** The smoothing parameters cannot be easily related to the gauge length. We went again through the equations to try to find a quantitative mathematical relationship. Due to the adopted formulation, no a clear direct relationship was found. However, a few considerations can be drawn by examining their definition and effects on the solutions. For the SIM, the more sensitive smoothing parameter is the distance factor introduced to avoid singularities in the analytical solutions. The smaller it is, the sharper the solution. For the SGM, the smoothing parameter D weights the relative contribution of the station with distance in the weighted least square inversion. The smaller it is, the lesser the influence of the station at larger distance. So both of them locally average displacement/velocity to derive strain. Their performances change with the signal wavelength. Smaller distance smoothing parameters are adapted for shorter wavelengths, whereas larger ones may perform better for longer wavelengths. It is worth of noting that the accuracy of the solution is dependent on the factor  $\frac{\sin(\pi L/\lambda)}{\pi L/\lambda}$ , that is the same multiplicative factor giving the DAS

response to axial strain over a gauge length of L (Bakku, 2015). This multiplicative factor comes, however, from two independent analyses based on the estimate of strain under spatially uniform approximation (Spudich and Fletcher, 2008) and on the computation of averaging strain over a gauge length L (Bakku, 2015).

**Anonymous Referee #1 (Specific comments):**

The average inter-station distance of the seismic array observation is approximately 70m that is larger than the gauge length of 10m. Therefore, it seems reasonable that the seismically estimated strain's spatial distribution is smoother than that of the DAS strain. I wonder if the authors can increase the gauge length by taking the spatial average of DAS strain records and compare that with the seismically estimated strain. I am interested to know to what degree such a smoothing averages out the small-scale medium heterogeneities

Authors: Thanks for giving us this hint. The DAS measurements were acquired at a fixed gauge length of 10 m. In the iDAS interrogator the user cannot change it. With sampling resolution of 25 cm, the iDAS internally uses an advance de-noising algorithm to improve the signal-to-noise ratio prior to data decimation to 2 m spacing. To properly estimate the effect of the gauge length it would have been ideal to have the raw data and set the gauge length before data decimation. However, following the suggestion of the reviewer, we attempt to investigate how the gauge length may average out the effect of the small-scale heterogeneities by virtually increasing the gauge length with a spatial average of DAS data. Results are included in the Supplementary Figures (Figs. S3-S8). We report the analysis on the VE event, where the effect of increasing the gauge length from 10 to 30 and 100 m is more significant due to the higher frequency content. Indeed, as expected, at higher gauge lengths the shorter wavelengths are filtered out (Fig. S3, S6). We used two averaging approaches: (i) a simple average over channels (Figs. S4, S7); (ii) a moving average with a shift of 1 channel (Figs. S5, S8). Also with a gauge length of 30 m, the simple average degrades the signals and the main phases are already lost. On the other hand, the moving average preserves the main signal but smooths out local scattering and reflections which are no more visible. When computing the misfit with the array derived strain estimates, the localized anomalies (Fig. 3) in coincidence of the faults due to the small-scale heterogeneities are flattened and broaden. These findings confirm that the distortion of the strain field is very localized and difficult to be observed by traditional seismic array methods, which require deployment of a dense network along well-chosen active faults and a good amount of luck. It is also important to stress that the fault system in Piano delle Concazze comprises minor faults in a complex volcano-tectonic setting. They mainly represent the accommodation in response to the extension of the North-East Rift. Being limited in extension and shallow (Napoli et al., 2021), distortions are expected to be narrow and very localized as our findings seem to support.

**Anonymous Referee #1 (Specific comments):**

It is necessary to show the correspondence between the location and the number of DAS channels. I would suggest showing the channel number in Figure 1 with a step of 100. Or the authors will be able to indicate the channel numbers at the corners and bends of the cable in Figure 1.

**Authors:** We appreciate this suggestion. Since Figure 1 is already too full of information, we indicate the channel numbers at the corners and bend of the cable in Figure 3 and 4.

**Anonymous Referee #1 (Specific comments):**

In equation (1), the second term on the top equation's right-hand side should be w, not p. And q, p, and w may be better to be explicitly shown even in this manuscript.

**Authors:** The term in the equation (1) was corrected. As suggested by the referee we added the expression of the Green's functions q, p, q as reported in Sandwell and Wessel (2016).

**Anonymous Referee #1 (Specific comments):**

On page 5, line 113, the authors mentioned that the authors used the least-squares method to solve the system. However, since the number of data and the number of unknown parameters are the same, the authors will solve the system directly. It is not necessary to use the least-squares method.

**Authors:** Thanks for highlighting this mis-information. We completely agree and we solved directly the system of linear equations. We corrected the text accordingly.

**Anonymous Referee #1 (Specific comments):**

In equation (5), the authors can show T (transposition operator) on the right-hand side, not on the left-hand side.

Authors: done.

**Anonymous Referee #1 (Specific comments):**

Page 6, line 155, fmax is about 7Hz, not 6Hz. Authors: done.

**Anonymous Referee #1 (Specific comments):**

In Figure 1, the authors need to mention the red circle at Bb04. Otherwise, I misunderstand that Bb04 malfunctioned.

**Authors:** To avoid misunderstanding, we changed the color of Bb04 like the other broadband seismometers. There is no particular reason to put Bb04 in evidence.

**Anonymous Referee #1 (Specific comments):**

At the end of the caption of Figure 2, an average scaling factor should be 1s/km, not 1000m/s.

**Authors:** We agree. In the text the scaling factor is reported as apparent slowness and not in apparent velocity. Accordingly, in the caption we report the scaling factor in apparent slowness, as suggested by the referee.

**Anonymous Referee #1 (Specific comments):**

In Figures 3 and 4, the authors just mentioned that the discrepancies are large around fault zones. However, it is necessary to mention clearly if the seismically estimated strain is overestimated or underestimated. **Authors:** The array derived strain estimates are in general underestimated in proximity of the fault. Amplification is observed in the DAS records as expected in presence of a damaged zone with lower velocity (Cao and Mavroeidis, 2019; Jousset et al., 2018).

**Anonymous Referee #1 (Specific comments):**

*I want to ask the authors to show Figures 7 and 8 on precisely the same scale as Figure 2.* **Authors:** The figures were redone to set precisely the same scale as Figure 2.

**On the comparison of strain measurements from fibre optics with dense seismometer array at Etna volcano (Italy)**

Gilda Currenti1, Philippe Jousset2, Rosalba Napoli1, Charlotte Krawczyk2,3, Michael Weber2,4

1Istituto Nazionale di Geofisica e Vulcanologia-Osservatorio Etneo, Piazza Roma 2, Catania, Italy

2GFZ German Research Centre for Geosciences, Telegrafenberg, Potsdam 14473, Germany

3Technical University Berlin, Ernst-Reuter-Platz 1, Berlin10587, Germany

4University Potsdam, Karl-Liebknecht-Str. 24-25, Potsdam 14476, Germany

Correspondence to: Gilda Currenti (gilda.currenti@ingv.it)

**Supplementary Information**

**Figure S1:** Time series (a, b) and spectra (c, d) during a small volcanic explosion (VE) at Etna on 6 July 2019. (a) Raw DAS strainrate; (b) strain computed by integrating over time the raw strainrate data; (c) spectra of the raw strainrate DAS records overall the channel; (d) spectra of strain records.

---

## Author Comment (AC2)

**Manuscript Ref. No. se-2020-216:** "*On the comparison of strain measurements from fibre optics with dense seismometer array at Etna volcano (Italy)*" by Gilda Currenti et al.

***Referee #2 (General Comments):***

*I enjoyed reading this manuscript. It's very well written and the figures are overall clear.*

*My only major issue concerns the general objective of the paper. In the abstract, you present your experiment as a way to understand DAS potential to detect volcano-seismic signal. You insist on the great opportunity to probe its multi-faceted activity which is again very promising. But in the end, you only analyze a volcanic explosion and a LP, and rather focus on the capability to measure strain by an in-depth comparison with the more conventional broadband deployment. This is fair and, as mentioned, interesting scientifically. Yet, I'm finding the abstract and the conclusion slightly misleading. It would be interesting to know if you have detected tremor for example? I fully understand the idea of focusing on the capability to retrieve strain measurements at volcanoes, but would modify the abstract, introduction and conclusion accordingly.*

*Overall, as mentioned above, I'm very positive and have only several minor comments listed below.*

**Authors:** We are pleased that the reviewer found our paper to be interesting and suitable for the journal. We would like to confirm that the main aim of the paper is to validate the DAS records. The title itself was properly chosen to highlight the main subject of the manuscript. Also the Introduction is almost dedicated to theoretical methods for indirect strain estimates and the proposal of an alternative methods of DAS records validation. For the first time we have the possibility to compare direct DAS strain with estimates of strain from a very dense seismic array at the summit of an active volcano. Since the cable was deployed in a scoria layer, we wonder if coupling was good enough to accurately sense strain generated by volcanic activity. The dataset is very rich due to the variety of volcanic activity of Etna. But, before analyzing the full dataset, we wanted to validate the measurements, which is the topic of this work. The single station procedure is the most method used for DAS data validation, which relates strain and particle velocity in the plane wave approximation. However, it suffers from the uncertainty on phase seismic velocity values, that regulate this relationship. Moreover, as pointed in the discussion, in volcanic region, where media are heterogeneous and possibly dispersive, making the estimate of the phase seismic velocity is highly challenging. Therefore, we looked for alternative methods which could give strain estimates without any assumption on the phase seismic velocity values. The good fit between DAS strain records and seismic derived strain provides us for the first time the "passport" that strain is properly recorded by DAS interrogator.

However, as suggested by the reviewer, to stress the main objective of the paper, we rephrased some sentences, where appropriate (e.g., in abstract and conclusion).

***Referee #2 (Specific Comments):*** *27: suggest adding a comma: 'Nowadays, advances'.*
**Authors:** done

***Referee #2 (Specific Comments):*** *27-30: suggest adding references to support these statements and guide the readers.*

**Authors:** We added further references and moved here the references, that were provided at the end of the paragraph (line 32-33 in the first version of the ms).

***Referee #2 (Specific Comments):*** *L.35: what do you mean by multifaceted?*
**Authors:** As often occur at Etna, the volcanic activity was characterized by a great variety of eruptive style going from effusive to explosive with different level of intensity. We better clarified it in the ms.

***Referee #2 (Specific Comments):*** *35: again some references are needed for Etna volcanic activity*
**Authors:** References have been added to describe the eruptive style of Etna volcano and the description of the activity during the acquisition period.

***Referee #2 (Specific Comments):*** *39: as a non-native English speaker, I'm not sure about measures. I have always used measurements instead of measures. Worth checking this in detail.*
**Authors:** "measures" was changed with "measurements".

***Referee #2 (Specific Comments):*** *46: 'constraints on seismic sources': could you be more specific here?*
**Authors:** References were checked and the content of the cited papers was better described.

***Referee #2 (Specific Comments):*** *52-53: 'close sensors or dense arrays': it would be good to give some numbers here, again to guide the non-seismologists who may be interested in your contribution*
**Authors:** Numbers and references were provided to give an outline of dense array deployments.

***Referee #2 (Specific Comments):*** *58: 'the Authors'-> 'the authors'*
**Authors:** done

***Referee #2 (Specific Comments):*** *L.90: what's the natural frequency of the Trillium Compact*
**Authors:** We used Trilium Compact – 120 s broadband sensors. We added this information in the ms.

***Referee #2 (Specific Comments):*** *115-116: how do you tune them?*
**Authors:** The smoothing parameters are tuned by performing a simple grid search with the aim to improve the fitting between observed and estimated strain. When no direct measurements of strain are available to compare, the tuning is performed on the deformation/velocity data, by omitting one sensor from the computation and then, after calculating the displacement/velocity at the same station, the recorded and interpolated displacements/velocities are compared (Paolucci and Smerzini, 2008).

***Referee #2 (Specific Comments):*** *L.125: odd : the size of the text suddenly increases*
**Authors:** The size of the font was adjusted to 10 overall the ms.

***Referee #2 (Specific Comments):*** *L.155: it would be interesting to take a similar gauge length for the DAS to more directly compare with the broadband array*

**Authors:** The used iDAS interrogator has a fixed gauge length of 10 m, that cannot be changed by the user. It would indeed have been interesting to perform measurements at different gauge lengths. Following the suggestion of the Referee #1 we performed additional computations to observe the smoothing of the DAS strain as gauge length is increased. We report the results in Supplementary Figures.

***Referee #2 (Specific Comments):*** *L.172: the font size has again changed for some reason*

**Authors:** The size of the font was adjusted to 10 overall in the text.

***Referee #2 (Specific Comments):*** *L.183: is that really giving a local estimate of their respective performance? I have the feeling it somewhat highlight their discrepancy but not their performance.*

**Authors:** We agree with this comment and accordingly changed the text.

***Referee #2 (Specific Comments):*** *L.189: comment on Poisson's ratio value: how did you come to this conclusion?*

**Authors:** We tested a range of Poisson ratio from -1 (fully decoupled) to 0.25 and 0.5 (elastic) to 1.0 (incompressible). By comparing the misfits, the solutions with an optimal value of 0.25 was chosen.

***Referee #2 (Specific Comments):*** *L.191: I only partly agree. RMSE misfits are also much higher in the northern and southwestern sections. Could you comment on this? But I agree with the fault zones.*

**Authors:** Thanks for highlighting this feature. Indeed, we are fully aware that larger misfits are along the two nearly EW branches (channel 134-302; 449-787), because the strain wavefield (Fig. 2 and Figs. S1-S2) is more complex and amplified with respect to the nearly NS branch (channel 302-449). This feature is maybe related to the relative direction between the main structural geology and the cable branches.

***Referee #2 (Specific Comments):*** *L.194-195: this sentence is not clear to me. What is your aim? To assess DAS performance? Why on irregular points?*

**Authors:** Usually, interpolation are performed on regular grid to compute displacements/velocity and then to derive strain/strain rate tensor by simply finite difference scheme. The derivation of analytical expression for strain allows for directly computing the strain on points distributed irregularly overcoming the limit to use regular grid. This also allows for a better accuracy. Since DAS cable layout are usually not regular in geometry, the proposed interpolation methods offer advantages with respect to simple gradient method. This concept is fully described at the end of the Discussion. To avoid repetition and misunderstanding due to a too short sentence at the beginning of the paragraph, we remove the L194-195.

***Referee #2 (Specific Comments):*** *The first paragraph is very convincing. Well done.*

**Authors:** Thanks for appreciating.

***Referee #2 (Other Questions):*** *207: I suggest using 'coinciding with' rather than 'in correspondence'*

**Authors:** done.

***Referee #2 (Other Questions):*** *L.220: have you tried to compute simple H/V ratios for the broadband to better understand the site effects?*

**Authors:** Thank for the suggestion. A study is undergoing to characterize the ground response in Piano delle Concazze using and comparing several methods. H/V ratios is one of the methods we are investigating.

***Referee #2 (Other Questions):*** *I would like to see a spectrum for each event before any filtering, perhaps as supplementary material.*

**Authors:** We added two Supplementary Figures (Figs. S1 and s2) which show the raw strainrate DAS records and its spectra for the two events. The spectra are computed and plotted all along the DAS channels. We also show the strain computed by integrating over time the DAS records. The signals are dominated by the continuous volcanic tremor at Etna whose seismic frequency band is in 0.5-5 Hz (Cannata et al., 2009). It is worth of noting the small amplitude strain changes related to the LP event, which is barely visible in the raw strainrate data.

***Referee #2 (Other Questions):*** *There seems to be more phases excited following the VE compared to the LP. Could you comment on this?*

**Authors:** Yes, it is true. As already noted during the experiment performed in 2018 (Currenti et al., 2020), the explosive events excite more phases due to scattering and reflection on faults and layered geology.

Currenti, G., Jousset, P., Chalari, A., Zuccarello, L., Napoli, R., Reinsch, T., and Krawczyk, C.: Fibre optic Distributed Acoustic Sensing of volcanic events at Mt Etna, EGU General Assembly 2020, Online, 4–8 May 2020, EGU2020-11641, https://doi.org/10.5194/egusphere-egu2020-11641, 2020

***Referee #2 (Other Questions):*** *Figure 1: strange geometry. Could you explain the rationale for selecting this geometry? What is the DEM resolution?*

**Authors:** The rationale behind defining the DAS geometry was guided by the aim to record dynamic strain changes along several directions. The Digital Surface Model has a resolution of 2 m (Palaseanu-Lovejoy et al., 2020).

***Referee #2 (Other Questions):*** *Figure 2: I would suggest to avoid the use of jet colours for the colormap (https://gorelik.net/2020/08/17/what-is-the-biggest-problem-of-the-jet-and-rainbow-color-maps-and-why-is-it-not-as-evil-as-i-thought/).*

**Authors:** We prefer to not change the colormap.

***Referee #2 (Other Questions):*** *I can't see the red line: 'DAS channel 501 (red line)'*

**Authors:** Thanks for noting this typo. There is no red line and we remove it in the caption.

***Referee #2 (Other Questions):*** *Figure 4: you should again mention here what are the open circles and black lines*

**Authors:** We added the meaning of open circles and black lines.

***Referee #2 (Other Questions):*** *Figure 6: the green color is very hard to see*

**Authors:** We changed the green color in dark green and increased the thickness of the lines.

***Referee #2 (Other Questions):*** *Figure 7: any reason why there seems to be more scattering for some channels, e.g., 200? Any local amplification?*

**Authors:** Channel 200 is in coincidence with the first fault that the optic fibre cable crosses along its path. Therefore, the scattering is likely associated to this fault. To better read the data and results, we show the channel numbers at cable corners in Figure 3 and 4.

***Referee #2 (Other Questions):*** *Figure 8: why are there some discrete lines around channels 600 and 300 for example?*

**Authors:** Around channel 600 and 300 the cable turns abruptly.

**On the comparison of strain measurements from fibre optics with dense seismometer array at Etna volcano (Italy)**

Gilda Currenti[1], Philippe Jousset[2], Rosalba Napoli[1], Charlotte Krawczyk[2,3], Michael Weber[2,4]

[1]Istituto Nazionale di Geofisica e Vulcanologia-Osservatorio Etneo, Piazza Roma 2, Catania, Italy
[2]GFZ German Research Centre for Geosciences, Telegrafenberg, Potsdam 14473, Germany
[3]Technical University Berlin, Ernst-Reuter-Platz 1, Berlin10587, Germany
[4]University Potsdam, Karl-Liebknecht-Str. 24-25, Potsdam 14476, Germany

*Correspondence to*: Gilda Currenti (gilda.currenti@ingv.it)

**Supplementary Information**

[Figure]

**Figure S1:** Time series (a, b) and spectra (c, d) during a small volcanic explosion (VE) at Etna on 6 July 2019. (a) Raw DAS strainrate; (b) strain computed by integrating over time the raw strainrate data; (c) spectra of the raw strainrate DAS records overall the channel; (d) spectra of strain records.

[Figure]

**Figure S2:** Time series (a, b) and spectra (c, d) during an LP event (LP) at Etna on 27 August 2019. (a) Raw DAS strainrate; (b) strain computed by integrating over time the raw strainrate data; (c) spectra of the raw strainrate DAS records overall the channel; (d) spectra of strain records.

[Figure]

**Figure S3:** Strain data (volcanic explosion on 6 July 2019) after increasing the gauge length to 30 m. Two methods were used: (top) averaging the data every 15 channels (30 m); (bottom) averaging the data with a mobile mean over 15 channels with a shift of 1 channel.

[Figure]

**Figure S4:** Misfits between the array derived strain and the DAS strain data (volcanic explosion on 6 July 2019) after increasing the gauge length to 30 m using a simple average.

[Figure]

**Figure S5:** Misfits between the array derived strain and the DAS strain data (volcanic explosion on 6 July 2019) after increasing the gauge length to 30 m using a moving average.

[Figure]

**Figure S6:** Strain data (volcanic explosion on 6 July 2019) after increasing the gauge length to 100 m. Two methods were used: (top) averaging the data every 50 channels (100 m); (bottom) averaging with a mobile mean over 50 channels with a shift of 1 channel.

[Figure]

**Figure S7:** Misfits between the array derived strain and the DAS strain data (volcanic explosion on 6 July 2019) after increasing the gauge length to 100 m using a simple average.

[Figure]

**Figure S8:** Misfits between the array derived strain and the DAS strain data (volcanic explosion on 6 July 2019) after increasing the gauge length to 100 m using a moving average.